# How Can Out-of-Hospital Cardiac Arrest (OHCA) Data Collection in Slovenia Be Improved?

**DOI:** 10.3390/medicina59061050

**Published:** 2023-05-30

**Authors:** Luka Petravić, Evgenija Burger, Urša Keše, Domen Kulovec, Rok Miklič, Eva Poljanšek, Gašper Tomšič, Tilen Pintarič, Miguel Faria Lopes, Ema Turnšek, Matej Strnad

**Affiliations:** 1Faculty of Medicine, University of Maribor, Taborska ulica 8, 2000 Maribor, Slovenia; rok.miklic@student.um.si (R.M.); eva.poljansek@student.um.si (E.P.); strnad.matej78@gmail.com (M.S.); 2Faculty of Mathematics and Physics, University of Ljubljana, Jadranska ulica 21, 1000 Ljubljana, Slovenia; burger.evgenija@gmail.com; 3Faculty of Computer and Information Science, University of Ljubljana, Večna pot 113, 1000 Ljubljana, Slovenia; ursa.kese@gmail.com; 4Faculty of Medicine, University of Ljubljana, Vrazov trg 2, 1000 Ljubljana, Slovenia; domen.kulovec@outlook.com; 5Faculty of Pharmacy, University of Ljubljana, Aškerčeva cesta 7, 1000 Ljubljana, Slovenia; gt1564@student.uni-lj.si; 6Faculty of Mechanical Engineering, University of Novo Mesto, Na Loko 2, 8000 Novo Mesto, Slovenia; tilen.pintaric99@gmail.com; 7Faculty of Electrical Engineering and Computer Science, University of Maribor, Koroška cesta 46, 2000 Maribor, Slovenia; miguelfarialopes2000@gmail.com; 8Faculty of Law, University of Maribor, Mladinska ulica 9, 2000 Maribor, Slovenia; ema.turnsek@student.um.si; 9Prehospital Unit, Emergency Medical Services, Community Healthcare Center Maribor, Ul. talcev 9, 2000 Maribor, Slovenia; 10Emergency Care Department, University Medical Center Maribor, Ljubljanska ul. 5, 2000 Maribor, Slovenia

**Keywords:** out-of-hospital cardiac arrest, Slovenia, registries, cardiopulmonary resuscitation, reference standards, emergency medical services, public health, data science

## Abstract

*Background and Objectives*: The prevalence of out-of-hospital cardiac arrest (OHCA) has been established as a significant contributor to mortality rates in developed nations. Due to the challenges associated with conducting controlled randomized trials, there exists a necessity for the collection of high-quality data to enhance the comprehension of the impact of interventions. Several nations have initiated efforts to gather information pertaining to out-of-hospital cardiac arrest (OHCA). The Republic of Slovenia has been collecting data from interventions; however, the variables and data attributes have not yet been standardized to comply with international standards. This lack of conformity poses a challenge in making comparisons or drawing inferences. The aim of this study is to identify how to better gather OHCA data in Slovenia. *Materials and methods*: The Utstein resuscitation registry protocol (UP) was compared to the Slovenian data points that must be gathered in accordance with the Rules on Emergency Medical Service (REMS) during interventions. In addition, we have proposed alternative measures of digitization to enhance pre-hospital data. *Results*: Missing data points and attribute mismatches were detected in Slovenia. Eight data points necessitated by the UP are gathered in several databases (hospitals, the National Institute of Public Health, dispatch services, intervention reports from first responders, and defibrillator files), but not in the mandated protocols based on REMS. Two data points have variables that do not match those of the UP. A total of 16 data points according to UP are currently not being collected in Slovenia. The advantages and potential drawbacks of digitizing emergency medical services have been discussed. *Conclusions*: The study has identified gaps in the methods employed for collecting data on OHCA in Slovenia. The assessment conducted serves as a basis for enhancing the process of data collection, integrating quality control measures across the nation, and establishing a nationwide registry for out-of-hospital cardiac arrest (OHCA) in Slovenia.

## 1. Introduction

Emergency medical services (EMS) in Slovenia are a part of the public health service network and are organized for the provision of pre-hospital emergency medical care and emergency transport of patients. EMS are regulated in accordance with the Rules on Emergency Medical Service (REMS) [1]. During each intervention, the EMS team fills out a protocol in accordance with the REMS the Emergency Intervention Protocol, and (in the case of resuscitation) the Pre-Hospital Resuscitation Protocol [1]. Data on the reception of calls from the dispatch center are collected separately as well as data on patients that are declared dead upon the arrival of EMS which are all collected using a separate form [2].

The collection of health data is a crucial first step in quality assurance, identifying trends, and improving health care [3]. The Slovenian Cancer Registry has a long history [4], but there is still no similar unified out-of-hospital cardiac arrest (OHCA) registry in Slovenia, although basic data have to be reported to the Ministry of Health [1]. 

The majority of European countries (Appendix A) have implemented a national OHCA or even a general cardiac arrest registry of some kind. Most of them are based on Utstein-style guidelines [5].

The Utstein resuscitation registry protocol (UP) from 2015 for OHCA serves as a guideline for the development of national data tracking programs. Standardization of data collection methods enables the comparison of the efficiency of EMS within the country itself as well as internationally [6]. Guidelines are provided for the collection of data on the system itself, dispatch service, patient data, resuscitation data, and treatment outcome [6], which are five pillars of OHCA patient care.

The aim of this study is to index the already collected data and define the opportunities for the improvement of data collection on OHCA in Slovenia. The main goal was to analyze which data on OHCA are gathered elsewhere and what are required by the UP [6]. The secondary goal was to elucidate good digitalization practices and make the reader aware of possible pitfalls.

## 2. Materials and Methods

Firstly, the indexing of data points that are followed in Slovenia and required according to the REMS was carried out [1]. Secondly, the list of national data requirements was compared to that of the UP [6]. REMS have been in effect since their approval in 2015 and are mandatory to be followed by all EMS providers in Slovenia. 

### EMS System in Slovenia

A call to 112 is always directed to the regional notification center from which it is directed to the health dispatch service (HDS) as long as the place of the incident falls under the catchment area of the EMS, which is included in the HDS. If the EMS is not included in the HDS, the call is redirected directly to the target EMS. Slovenia is still in the stage of setting up a HDS that will cover all of the geographical areas of the country.

Care for emergencies in the out-of-hospital setting is provided by advanced life support (ALS) teams and basic life support (BLS) teams. Each ALS team includes an emergency medicine-trained physician providing ALS and BLS teams comprising a registered nurse and/or medical technician. Upon receiving an emergency call with suspicion of OHCA, the ALS team is dispatched to the scene. If OHCA is not suspected and the patient does not need immediate critical care, BLS is dispatched. In a scenario of the ALS team being busy with another case, the BLS team is dispatched and the ALS team joins later.

In 2014, specially trained first responders were introduced into rural areas of Slovenia, with a response time of more than 10 min [7]. Most of them are operational volunteer firefighters. According to the REMS 2015, they should possess the knowledge and skills for the resuscitation of adults and children using AEDs, treatment of airway obstruction due to a foreign body, and treatment of major isolated external traumatic bleeding. After the training program, they receive a license, which is to be renewed every year [1]. 

Activation of first responders has been shown to shorten the time from the call to the initiation of CPR and the first defibrillation in Slovenian remote locations as well, which is in agreement with the findings of studies conducted abroad [8].

## 3. Results

### Comparison between UP and Data Being Gathered in Slovenia

When comparing data collected in Slovenia to that under the UP, the data can be divided into five major categories: elements that comply with the UP and are collected under mandatory protocols, elements that comply with the UP but are collected elsewhere (Table 1), elements that partially comply with the UP but have variables that do not match those of the UP (Table 2), and elements that are missing (Table 3). 

Partially missing elements, as shown in Table 2, are those whose values only partially overlap with those of the UP. This means that there are several alternative value possibilities, and only a small number of these options match the ones required under the UP.

Table 3 represents missing elements together with the values that are expected under the UP. When it comes to identifying missing data, careful research of all possible data sources is necessary. 

Core aspects are the number of cardiac arrests attended, resuscitation attempts made, and resuscitation attempts not made. It is worth noting that the majority of the other items in the table are defined as supplementary.

## 4. Discussion

Data points that are already being collected as mandated by the Slovenian legislation (REMS) cover most of the data points required by the UP. The remainder of the data already collected in Slovenia may be obtained from hospitals, the National Institute of Public Health, dispatch services, and intervention protocols from first responders—a total of four sources that vary depending on the local municipality. In Slovenia, access to this data is currently difficult since data on all the OHCAs attended by the EMS are not collected in one location, but rather are fragmented between municipalities. Data are also not machine-readable or recorded uniformly, making automation infeasible.

OHCA remains an important and substantially underestimated public health problem in both Slovenia and Europe [9]. According to EuReCa TWO research, there are substantial disparities in survival rates between nations with a larger percentage of patients who had received CPR, than that of those who had not [10]. Efforts are necessary to set up systems that would help understand how to improve the result both locally and internationally. Since randomized control trials are uncommon for this group of patients, researchers have to rely on evidence from gathered OHCA registries to understand the nature of this epidemic [11].

The United States CARES registry, which includes 140 million people in 28 states, shows that a high-quality nationwide OHCA register is possible. Furthermore, it has exposed the OHCA burden, which goes beyond mere survival and mortality statistics. Without a quality data framework in place, it is inevitable that crucial trends may be overlooked. There is no registry like CARES in Slovenia or Europe [9].

The UP standard is widespread across multiple registries around the world, which makes data comparison easier. Because it is endorsed by the European Resuscitation Council, it is currently the standard for European OHCA research [6].

### 4.1. Challenges Regarding Data Collection in Slovenia

There are challenges in the Slovenian EMS reporting system that should be harmonized. In addition to the data that are not being collected at all, there are several variable attributes that are not completely compatible with the UP [6]. Since there is already a well-founded opinion that a joint European registry should exist to best address the impact of cardiovascular events [9], the UP should be implemented from the ground up as it represents an internationally recognized reporting protocol [6]. To achieve that, data points of the EMS protocols in Slovenia should be amended.

EMS workers in Slovenia use defibrillators to track vital signs, and additionally, some AEDs also have a memory function that enable the export the of data. To streamline the reporting process, it is essential to index data automatically. The problem is proprietary formats that need licensed software to be read, making it impossible to access them without increasing the budget of the end user’s organization or department. This could be overcome with an international standard that is required by law in the European Union, making the raw data (that is owned by the patient) useful for improving patient care and outcomes [12]. Exported files should be converted into a machine-readable data format and automatically synced with the database. The data which could be saved in these formats include time codes, initial heart rhythm, capnometry, oxygen saturation, energy delivered during defibrillation, etc. This can be at this time in most parts of Slovenia exported only in a format which is not machine-readable and therefore not useful in the case of big data analytics.

In Slovenia, data on cases in which resuscitation was not attempted is currently not being collected. It is important to include those cases as well as the OHCA events where the patient was transported to the hospital using private transportation or other means [5].

### 4.2. Slovenian Compliance with the UP

It is crucial to highlight that there were some uncertainties discovered while examining the available data. For example, targeted temperature management (TTM); the UP specifies the time and location where TTM begun, whereas Slovenian standards merely gather information in the form of free text on how TTM was carried out. This implies that information on TTM is gathered, and in certain situations, the supplied value may match that required under the UP; however, this is unstructured and therefore unusable on a big scale.

Despite some deviations and challenges, most of the data points required under the UP standards are already being collected in Slovenia as already required by the Slovenian legislation [1]. Adding the missing data points to the mandatory protocols would therefore not represent a significant additional burden to healthcare workers who are responsible for the documentation of each OHCA case.

It is also necessary to distinguish between the items obtained under protocols and the handful of elements collected elsewhere. A detailed examination of these characteristics might be useful in determining how to begin adding missing data into the existing system. This is required to avoid burdening EMS professionals with too many additional procedural responsibilities when there are already partial responses available, notably for supplementary items.

When the data gathering process starts, several potential issues are anticipated. For example, each institution has its unique method for storing sensitive information, making it difficult to collect and access. Furthermore, while gathering data from several sources, it is inevitable that the data will be inconsistent [13]. To compute defibrillation time, for example, the time of the first shock given must be retrieved from the defibrillator. The clocks of separate institutions, or even different computers within the same institution, are not synchronized, making time computations unnecessarily less accurate [14]. Addressing this entails system-wide digitalization with clearly set standards for both software and hardware.

### 4.3. The Effect of OHCA Registries Abroad

There is already an existing functional and high-quality national registry in the U.S. (CARES). However, in Europe, there is a need to establish an international registry [9]. Numerous countries where an OHCA registry has been implemented report a significant improvement in clinical outcomes. It is important to note that the registry in itself does not improve survival, but rather actions taken by the system and users do. The registry only helps in realizing the actual state, acting upon it and following the outcomes. Changes in guidelines that were made directly because of the CARES registry resulted in a 73% improvement in overall survival in Detroit over 3 years. The survival rate increased from 3.7% in 2014 to 6.4% in 2016 [15]. The SCAR (Swedish Cardiac Arrest Register) is also believed to have significantly contributed to the improved OHCA survival rates in Sweden [16].

Based on the data gathered in the Danish OHCA registry, it was possible to identity a small number of bystanders and low 30-day survival as some of the weak points of the Danish system. Based on the data, it was possible to develop a national strategy to optimize the system. As a result of that, it was possible to improve the rates of bystander CPR and survival; however, during that period, multiple factors in the system improved, which makes it impossible to draw direct correlations between bystander CPR and survival [17]. The All Japan Utstein Registry also indicates a positive survival trend over time [18].

OHCA registries have been proven useful in strengthening the chain of survival around the world which involves American Heart Association-identified measurement, benchmarking, and providing feedback, as identified as essential elements of quality resuscitation systems in 2010 [18]. On the other hand, it is important to emphasize the multifactorial nature of OHCA outcomes and the limitations of registry data in determining the exact factors that contribute to favorable outcomes [18].

### 4.4. Utilization of Defibrillator and AED Data in Slovenia

Defibrillators and AEDs capture many data points that could be used for further analysis, observational studies, and even machine learning (ML) training models. At the moment, these data are not being systematically collected or analyzed in Slovenia. Technological solutions that could be used for the gathering of the data and connecting it to a central server already exist and could be used to improve both patient care and research.

The technology that sends data from defibrillators already exists [19,20]. To make OHCA data collection in Slovenia more comprehensive we propose that all defibrillators and AEDs receive required hardware upgrades to enable software connection. Defibrillator data should be imported into the OHCA registry and connected to patients’ unique identifiers.

### 4.5. Digitalization of Protocol Intake

Slovenian EMS practitioners are already obliged to report, in addition to other interventions, every OHCA intervention as mandated by the law. Currently, the data are being collected on paper forms [1]. We suggest digitizing the forms to streamline data gathering and standardize the user experience.

The implementation of digitalization in everyday work routines has strengths and weaknesses which have to be taken into account in order to best implement new technologies and give both the patients and healthcare professionals the best user experience possible. A report on the project of digitizing the NHS found that digitalizaiton efforts are not always prudent, as it cites a failed attempt at NHS ambulance dispatch service digitalization in 1993 and failed national IT digitalization in 2013 [21]. The report also puts forward the need to listen to the end users as digital EHRs can require the attending physicians to make well over 4000 mouse clicks in a 10 h shift [21]. However, there are also upsides: with good implementation, digitalization makes it easier to share data across the medical team, making it easier to communicate important patient information to the emergency department before the arrival of the patient, giving paramedics support with decisions in the field and enabling the secondary use of data for research purposes [21]. Introducing EHRs into a pre-hospital environment has another strong advantage, as it lowers the chances of mistakes [22]. Electronic hand-off papers are strongly preferred over hand-written ones, especially because of their readability, as handwriting can pose a significant problem in conveying information [22]. Younger generations of paramedics report an easier change from paper and pen to digital technologies than older members of the team do [22]. Paramedics who have implemented EHRs into their practice find that the data are more objective and significantly more accurate [22]. A qualitative study in the ED department found that EHR documentation time increased five-fold compared to paper documentation, work roles were obscured with residents conducting more work than attendings compared to before and using paper documentation as an aid to transfer information from the patient’s bedside to the charting room [23]. On the contrary, the installation of full electronic medical records (EMRs) reduced the time a patient spent in the emergency department by 22.4% and diagnosis/treatment time by 13.1%, according to a study including 364 hospitals in the United States. Time optimization in emergency rooms is particularly important in view of Slovenia’s growing dearth of emergency medicine specialists [24].

In Slovenia, EMS digitalization has a long way to go. Currently, only 10 of the 18 EMS departments can electronically input patient data forms. In addition, not all of them are affiliated with the national dispatch center. Another issue that must be addressed is the variety of software they employ—18 departments reported using four distinct IT service providers [25]. In addition to software, adequate hardware must also be taken into account.

The issue of digitizing EMRs in ambulances is not unique to Slovenia. Despite the widespread belief among EMS employees that digitalization would improve their efficiency, there are international reports of insufficient digitalization with hurdles comparable to those in Slovenia (stability, hardware, functionality, and compatibility with external entities, such as hospitals and dispatch centers) [26,27].

### 4.6. Healthcare Digitalization at the National Scale

EMS are a component of a bigger national healthcare system. To ensure interoperability, the best usability, and the highest data standards, digitization must be meticulously planned and handled thoroughly.

The majority of healthcare facilities in Slovenia employ electronic health records (EHRs), yet the IT solutions often entail more work, not less. The European Commission ranked the Slovenian solution for unified national healthcare data collection (eZdravje) sixth in the Digital Economy and Society Index in 2019. However, users of internal IT systems in hospitals report that data entry is slow and that the same data must frequently be entered multiple times, which is burdensome for all parties involved [28].

Digitalization has been effective at facilitating higher-quality and safer patient care [28]. A shift toward a value-based healthcare system that prioritizes the patient will further demonstrate the utility of data in ensuring superior patient care. The Medical Chamber of Slovenia has already developed principles for a value-based healthcare system in Slovenia, and high-quality digital databases are essential for its implementation [3].

The Slovenian Ministry of Health published a national digitalization strategy in November 2022, with objectives to be met by 2027. Regarding patient identification and connecting different data gathered, the strategy proposes the use of a unique patient ID that is similar to the patient ID used in the picture archiving and communication system (PACS) used in radiology. This makes it safer, in personal identifiable data are not used directly (i.e., social security number, name and surname or date of birth, but rather pseudonymized. To maintain interoperability and a uniform EMR throughout all phases of patient care, it is vital that all healthcare IT developers adhere to it. We must ensure that an OHCA registry is a component of this larger system and not a stand-alone database—a data lake, not a data silo.

### 4.7. Limitations

Our study is limited by the data protocols being compared applying only to Slovenian geographic area. Other countries might have different problems with the data being harmonized to international standards or none at all if the protocols are already harmonized to the latest version of the UP. We may not have identified or discussed all of the positive and negative effects of OHCA registries in other nations. This article focuses predominantly on the digitalization of OHCA data, but the Slovenian healthcare system could benefit from a more comprehensive digitalization process that encompasses a variety of healthcare issues.

## 5. Conclusions

The usefulness of data currently gathered in Slovenia was highlighted in this study. Slovenia must first comply with international standards in order to create a high-quality dataset of OHCA data and improve patient care. Slovenia already has a comprehensive system in place for tracking OHCA patient data, which are in most instances incompatible with the UP. The main issues are attribute mismatches and missing variables. When thinking about digitalization, it is very important to not focus only on the software, but also the hardware. Medical workers in hospital settings are surrounded by medical devices that already enable the sharing of data and that represent an untapped potential. Most of the groundwork has already been carried out. It now has to be harmonized, connected and used in a suitable manner. This comparative evaluation provides a groundwork for improving data models in Slovenia.

## Figures and Tables

**Table 1 medicina-59-01050-t001:** List of data defined by the UP and derived from sources other than the intervention protocol.

Data Not in Protocols	Values	Additional Data Source
Comorbidities	Yes/no/unknown/not recorded	GP, hospital
Defibrillation time *	mm:ss/unknown/not recorded	Defibrillator, Protocol from first responders
CPR quality	Yes/no/unknown/not recorded	Defibrillator
Number of shocks	Number/unknown/not recorded	Defibrillator, Protocol from first responders
Reperfusion attempted *	Typea Angiography only/PCI/thrombolysis/none/unknown/not recorded	hospital
ECLS	Before ROSC/after ROSC/not used/unknown/not recorded	hospital
Number and type of neuro prognostic tests	Recorded EEG—yes/no/unknown/not data from hospital	hospital
30 d survival or survival to discharge *	Yes/no/unknown/not recorded	GP, hospital

Note: * Core element is a data point representing the minimum recommended standard to follow under UP; GP—general practitioner, CPR—cardiopulmonary resuscitation, PCI—percutaneous intervention, ECLS—extracorporeal life support, ROSC—return of spontaneous circulation, EEG—electroencephalography, 30 d—30 days.

**Table 2 medicina-59-01050-t002:** List of partly missing data elements defined by UP.

Partly Missing Data	Values Required	Data Source	Data Collected in Slovenia
Airway control (type)	None used/oropharyngeal airway/supraglottic airway/endotracheal tube/surgical airway/multiple/unknown/not recorded	Pre-hospital resuscitation protocol	Cricothyrotomy/ alternative airway: i-gel/airtraq/laryngeal mask/laryngeal tube/conicotomy/tracheostomy/other
Vascular access (type)	Central line/peripheral IV/IO/endotracheal/unknown/not recorded	Emergency transport report	Venous route: no/failed/one/more than one
		Pre-hospital resuscitation protocol	Venous route: central line/IO

Notes: IV—intravascular, IO—intraosseous.

**Table 3 medicina-59-01050-t003:** List of missing data elements in prescribed protocols in Slovenia defined by UP.

Missing Data	Values	Current Location of the Data
Number of cardiac arrests attended *	Number of cases	Ministry of Health, annual report
Resuscitation attempted *	Number of cases	Ministry of Health, annual report
Resuscitation not attempted *	Number of cases	Ministry of Health, annual report
Independent living	Yes/no/unknown/not recorded	Social services/general practitioner
Targeted oxygenation/ventilation	O_2_ and CO_2_/O_2_ only/CO_2_ only/not used/unknown/not recorded. If this variable is reported, include details of specific targets in the system description.	Intensive care unit documentation
Reperfusion attempted *	Timing: intra-arrest/within 24 h of ROSC/>24 h but before discharge/unknown/not recorded	Intensive care unit documentation
IABP	Yes/no/unknown/not recorded	Intensive care unit documentation
pH	pH value/unknown/not recorded	Intensive care unit documentation
Lactate	Lactate value (mmol/L)/unknown/not recorded	Intensive care unit documentation
Number and type of neuro prognostic tests	SSEP—yes/no/unknown/not recorded, NSE—yes/no/unknown/not recorded, CT of the brain—yes/no/unknown/not recorded, MRI of the brain—yes/no/unknown/not recorded, clinical examination—yes/no/unknown/not recorded, other (define)—yes/no/unknown/not recorded. Indicate the timing of the test and whether or not the test led to discontinuation of treatment.	Intensive care unit documentation/general practitioner
Hospital volume	Number of cases/n **	Yearly hospital report
Targeted blood pressure management	mm Hg/no target set/unknown/not recorded	Intensive care unit documentation
Treatment withdrawn (including timing)	Yes/no/unknown/not recorded. Days/hours	Intensive care unit documentation
Organ donation	Number of cases/unknown/not recorded ***	Intensive care unit documentation
Patient-reported outcome measures (outcomes selected by patients as important)	Free text	Not collected at all
Quality of life measurements (standardized questionnaires, e.g., EQ-5D and SF-12)	Yes/no/unknown. List quality of life instrument(s) used and outcomes/scores.	Not collected at all

Notes: * part of regular reports as dictated by Rules on Emergency Medical Service; ** hospital volume is published in yearly reports, which are not published uniformly and are not machine-readable; *** organ donation has its own registry as Slovenia is part of EuroTransplant—slovenija-transplant.si; ROSC—the return of spontaneous circulation, n—no, SSEP—somatosensory evoked potential, MRI—magnetic resonance imaging, NSE—neuron-specific enolase, CT—computed tomography scan, IABP—intra-aortic balloon pump therapy.

## Data Availability

Not applicable.

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
