# Peer review of "How Can Out-of-Hospital Cardiac Arrest (OHCA) Data Collection in Slovenia Be Improved?"

_medicina, 2023, doi:10.3390/medicina59061050_

Round 1

Reviewer 1 Report

General comments:

I understand that this manuscript revealed the difficulties regarding OHCA data collection in Slovenia.

As a scientific manuscript, authors need to show the methodologies regarding data gathering and constructing the dataset according to Utstein guidelines from different organizations. I assume that there must be an issue regarding the identification of each dataset. Is it possible to use the patients” name, social security numbers or date of birth to connect each dataset?

If not, what kind of means do the investigators have in mind?

The tables show the data elements from different institutions. It would be better with the information regarding the data format. Some of them seem to be hand-written documents. It would be hard to change those datasets into electronic formats.

Authors discussed the impact of UP formatted datasets analysis on the improvement of OHCA outcomes. However, this seems to be outside of the study scope. 

Reviewer 2 Report

Study comparing the Utstein protocol for reporting OHCA with the present day of OHCA reports in Slovenia. This study found important potential improvements for the Slovenian system and is thus important on an international level for countries which have similar aspirations. The study is well conceptualised and written. Few typos, e.g. in Table 2 it should read Airtraq. Minor English improvements possible. 

Specific comment: Suggest to weaken the statement the outcome of OHCA improvement because of registries. Actually, the registry helps to record outcome and to assess changes in the medical approach and outcome over time

See above

Round 2

Reviewer 1 Report

Authors revised this manuscript well.